# Can Earth’s Mightiest Heroes Help Children Be Physically Active? Exploring the Immersive Qualities of Les Mills’ and Marvel’s “Move Like the Avengers” Video

**DOI:** 10.3390/ijerph18137184

**Published:** 2021-07-05

**Authors:** Emily Budzynski-Seymour, Michelle Jones, James Steele

**Affiliations:** 1Faculty of Sport Health and Social Sciences, Solent University, Southampton SO14 0YN, Hampshire, UK; james.steele@solent.ac.uk; 2Sport, Physical Education and Coaching, Plymouth Marjon University, Plymouth PL6 8BH, Devon, UK; mjones@marjon.ac.uk

**Keywords:** physical activity, children, health promotion, affect, enjoyment, music, characters, narrative

## Abstract

There have been calls for more enjoyable Physical Activity (PA) interventions which focus on ensuring a positive affective response. This study explored how using a narrative, characters, and music in a video-led PA session might influence the sense of immersion and impact the affective response. One hundred and thirty-six participants (boys n = 65, girls n = 71) were recruited, 85% aged between 7 and 11 years old. Participants completed the “Move Like the Avengers” PA video created by Les Mills and Marvel, then complete a survey answering questions on their post activity affective responses, and the use of immersive elements. Positive average affective responses were found (valence mean score: 3.6 ± 2.2, arousal mean score: 5.1 ± 1.0). Analysis revealed the narrative with characters indirectly mediate the valence response through creating a sense of immersion (β_std_ = 0.122 [95%CI 0.013 to 0.231]; *p* = 0.012). Musical elements had both a direct (β_std_ = 0.449 [95%CI 0.264 to 0.634]; *p* < 0.001), and an indirect (β_std_ = 0.122 [95%CI 0.014 to 0.229]; *p* = 0.011) effect upon valence and a direct effect upon arousal (β_std_ = 0.244 [95%CI 0.006 to 0.482]; *p* = 0.021). These promising results provide justification for future research into children’s immersive PA.

## 1. Introduction

Promoting physical activity to children is a high public health priority [1] and governments are increasingly prioritizing strategies to increase physical activity among young people [2]. Consistently evidence suggests that children are moving less and failing to meet the recommended levels of physical activity [3]. It has been suggested that enjoyment is a key factor for children’s engagement in physical activity, and as such recently there has been a call for more child based physical activity interventions to focus on ensuring an enjoyable experience [4]. This could be effective as feelings of enjoyment are strong motivators for physical activity in children [5,6,7]. This is also represented by a recent shift towards a more holistic approach to physical activity, taking into consideration the more affective and emotional components that may influence engagement [8]. A positive affective response is associated with increased engagement in physical activity, and it can help to build and sustain motivation for physical activity over time [9]. This suggests that targeting the affective and emotional responses to an activity could be an effective strategy for future physical activity interventions.

Despite the evidence for the importance of ensuring an enjoyable and fun physical activity experience for children, there are very few investigations into how to actually achieve this. Thus, this remains an under-researched area. Indeed, it has been suggested that a deeper understanding of the concept of fun is needed [10,11] and that research into what specific factors contribute to fun in physical activity is important considering this dearth in the literature. Most of the current research in this area is descriptive, for example Hopple [12] presented a top ten list of reasons why children find physical activity fun. Included in the list was that they enjoyed being “caught up in the moment” and “only concerned with the activity and nothing else”. This is interesting as it suggests that children like to be immersed in physical activity as they find this fun.

There are a number of methods that can be employed to create a sense of immersion during a physical activity session. Firstly, through the incorporation of a story or narrative. Narratives can promote perceptions of physical activity as fun [13] and are commonly used to promote health as they are a key persuasion tool due to their immersive qualities [14,15]. Narratives can transport people to another world, and they have the ability to influence attitude through a journey like experience [16]. When those engaged in the narrative are immersed this can suspend disbelief and reduce counterarguments and can create deep affection and affective responses [16]. Characters are also often incorporated into the narrative, those immersed in the story can identify with a character which can influence either the attitudes and/or behaviours of those immersed [16]. The use of characters also links into the research around wishful identification, which occurs when the viewer wants to be like the character, they want to actively emulate the figure, and the character is someone to whom they look up to [17]. This suggests that characters are potentially perfectly placed to help encourage physical activity in children. The characters can be incorporated into a narrative, so the children are immersed in the activity, and then as the characters engage in physical activity so will the children.

One additional component which can be used to help create a sense of immersion is the inclusion of music. Music can be used as a background sound, to direct attention towards something particular, as a warning sign, as an indication of positive or negative emotion, or as an acknowledgment of success [18]. Music can add to the immersive atmosphere, and is commonly incorporated in a physical activity session to aid in the control of psychomotor arousal and the regulation of affective states as it can help induce certain emotions, including happiness [19]. The inclusion of music in a child’s physical activity setting has been investigated previously [20,21,22,23] and has been shown to be a favourable addition for both affective and behavioural outcomes. However, the combined use of music, a narrative and characters has not yet been investigated. Furthermore, the ability of these elements to create a sense of immersion in children during their activity, and subsequent influence over their affective response to the activity experience has yet to be investigated. This research could potentially further the knowledge base around more enjoyable or fun activities for children and offer an insight into the use of more immersive activities for children.

There are existing resources available that do incorporate all of these elements and offer the opportunity to investigate their influence over children’s physical activity engagement in ecologically valid settings. One such resource is the online platform from Les Mills for children called Born to Move, which is full of physical activity videos. The philosophy of Born to Move is to create experiences where children can express themselves physically and develop a lifelong love of movement. In many of their classes they use fun and simple movements where children go on a journey to become characters in different immersive environments, with all their classes being set to music [24]. Having a fun and enjoyable experience is a key intended outcome [24]. In a recent video they teamed up with Marvel to create the “Move like the Avengers” session, where the instructors in the video take children through a series of “Avengers” themed moves, incorporating live action, animated graphics, sound effects, and cool music [25]. Marvel characters such as Thor and Black Widow make an appearance, and there are visual additions to the video which are aimed at increasing the immersion in the “Marvel world”. Figure 1 is a still taken from the video showing the characters and the instructors.

As this video incorporates a narrative, characters, and music, with an overall aim of ensuring immersion and fun, it was used in this research to explore the influence of these elements on children’s physical activity. The aim of the research was to explore the influence of these elements upon post activity affective responses, and whether effects were mediated by a feeling of immersion in children participating in the Move like the Avengers physical activity video experience.

## 2. Materials and Method

### 2.1. Study Design

An observational study design was used for this research. Ethical approval was gained from the lead authors institution, Solent University, Southampton, UK. The study was initiated in July 2020 with an initial sample recruited. This was then used to determine sample size for the main study (see below) and this was pre-registered on Open Science Framework in March 2021 (https://osf.io/4avtc, accessed on 30 June 2021) and all materials, code and data are available on the project page (https://osf.io/af4ew/, accessed on 30 June 2021).

### 2.2. Power and Sample Estimate

Participants were recruited in a sequential opportunity sampling procedure. Initially we recruited 85 participants whose data were analysed (preliminary results available in online materials https://osf.io/2u8ed/, accessed on 30 June 2021) prior to then conducting simulations to determine sample sizes required for our main analysis. We used simulation to determine sample size required for a priori power of β_power_ = 0.8, and an adjusted α = 0.05/2 = 0.025 to account for the initial ‘peek’ at the preliminary sample noted above. Our proposed exploratory partial mediation models (described in the Statistical Analysis section below, and in the online supplementary code along with our simulations https://osf.io/z4e5b/, accessed on 30 June 2021) was to examine the direct and indirect effects of (1) characters and narrative, and (2) music and audio-visual elements, upon affective response (valence and arousal). Population models were determined with moderate (β_std_ = 0.5), and also small (β_std_ = 0.25), standardised path coefficients. Samples (n = 1000 simulations) were then drawn from this population model across a range of sample sizes and power curves plotted (see https://osf.io/tk5db/, accessed on 30 June 2021). Based on this simulation a sample size of ~50 participants gave 80% power to detect moderate effects at an adjusted alpha of 0.025, and ~150 participants for small effects. Thus, we aimed to recruit up to ~150 participants. The stopping rule for this study was determined by time-constraints due to it forming part of PhD studies. Recruitment aimed for the sample size indicated but was stopped May 2021.

### 2.3. Participants

The final participant sample was 136 meaning that we had very high power (β_power_ = ~99%) to detect moderate standardised effects, but not quite sufficient a priori power for small standardised effects (β_power_ = ~70%).

In the final sample of 136 participants both males (n = 65) and females (n = 71) were included. Participants were recruited worldwide, including from the United Kingdom (n = 108), the United States of America (n = 13), Oceania (n = 3), European countries other than the United Kingdom (n = 10), South America (n = 1) and Asia (n = 1). 18 participants were aged six or younger, 116 were aged between seven and 11, and two were aged 12 or older. The majority of the children included did not have Les Mills membership within their household (106 = No membership, 30 = Membership). Table 1 details the participants characteristics for the included sample.

### 2.4. Recruitment

Recruitment was conducted through two separate pathways. Firstly, to target children at the beginning of the data collection (July 2020) social media posts were used to ask parents of children of the relevant age to consider their child taking part. This was done as the majority of schools were shut due to COVID-19 lockdowns. Additionally, a pop-up link was added to the online version of the video on the Les Mills website. This meant that whenever someone watched the video they would then be asked if they would answer some questions on their experiences doing so. They were then directed to the online survey. Just over half (51.5%) of the sample completed the session at home. In March 2021 when schools re-opened, two schools were then contacted to ask if they would help out with the research by playing the video during school PE time and getting the children to complete the survey. 48.5% of the sample completed the video at school. Ninety-three of the included participants reported that they had previously completed the video, however they had not completed the survey after the video before.

### 2.5. Procedure and Measures

In all cases children were asked to complete the video first, and then proceed to the online survey immediately after. The survey started with the participant information, this was spilt into information for the parents/guardians and then information for the children. Both were then asked to provide consent, if this was not given then no further progress could be made into the survey. When the survey was administered at school, prior to the completion of the video the school sent out a separate information sheet and informed consent form for parents, this informed them of the planned activity before the data collection. They were also told that an adult could help them with the survey, in reading and understanding the questions if needed. The survey (which can be found in the Appendix A) started by collecting some basic demographic information from the sample, including age, gender and location. Following this there were some questions on the environment where they did the session, e.g., at home or at school, and then some questions in regard to their previous exposure to the Avenger characters. A slight majority (58.1%) of the sample had previously watched an Avengers film and 67.6% of the sample reported that they had previously watched a film/TV show with Avengers characters in it or read any of the comic books where characters are present. Due to the timing of the data collection, being during a global pandemic, it was decided that some additional questions should be included to explore this descriptively in the sample. The children were asked both how happy and how worried they felt the previous day, and whether their physical activity levels had changed since the pandemic.

From this point the survey was split into four sections. Firstly, questions asking about the post activity affective response. The children’s feeling scale (CFS) and children’s felt arousal scale (CFAS) which have been previously adapted from the validated adult version [26,27] were used. There are only minor changes between the two versions, these being changes to the descriptors and the use of faces. Due to the limited research into collecting children’s affective responses, these measures were chosen as they have good face validity, they are conceptually easy for children to understand and have been used in a similar population in previous research [27,28]. Additionally, post activity affective responses have been shown to moderate the intention-behaviour relationship in an adult population [9,29,30]. The next section focused on the character and narrative elements of the video; three Wishful Identification questions were adapted from Hoffner [31] which have been used in a similar age group in previous research [32,33]. For the music questions three questions were adapted from the Brunel Music Rating Inventory [34], these were the questions relating to the speed of the music and how motivating children felt this was, the association factor of the music and the Avengers, and finally whether having the music helped them to exercise for longer. This inventory has been found to be valid in an adult population [34]. The final section related to the immersive qualities of the video and was adapted from the Immersive Experience Questionnaire, which has been previously validated and shown to measure cognitive involvement, real world dissociation, challenge, emotional involvement and control [35]. This questionnaire was developed from previous studies into the related areas of flow, cognitive absorption and presence [35]. The CFAS, CFS and Wishful Identification questions have all been previously used in research with children, and although the Immersive Experience Questionnaire and Brunel Music Rating inventory have only been validated and used in an adult population, there were adapted for use in this exploratory research study, and the choice was made to include them as they were thought to be conceptually easy and simple for children.

### 2.6. Statistical Analysis

Our pre-registered main analysis (see https://osf.io/4avtc, accessed on 30 June 2021), though framed within a null hypothesis significance testing framework, was considered exploratory. Mean scores for items for each construct were calculated for use as dependent variables in statistical analysis. As noted, we did not conduct our own psychometric analysis of the properties of the adapted scales used. Thus, the implied model for our constructs was a parallel factor model whereby unstandardised loadings and error variances were assumed identical across items [36]. For the present exploratory work, we considered this application of what is referred to as Intuitive Test Theory [37] as acceptable. Structural equation modelling (SEM) was used to analyse the data using the R package ‘lavaan’ [38]. Exploratory partial mediation models were applied to examine the direct and indirect effects of characters and narrative, and music and audio-visual elements upon children’s affective responses (valence and arousal). The specified model can be seen in the supplementary code (https://osf.io/ndh5y/, accessed on 30 June 2021). As noted in the pre-registration, preliminary analysis had been conducted on a sample of data and thus we applied an adjusted alpha of 0.025 for this main analysis to account for the initial ‘peek’ at the data made with the preliminary analysis. Model parameter estimates are reported as standardised path coefficients. We also report adjusted 97.5% confidence interval (CI) estimates to describe the precision of the estimates. Path diagrams for SEMs are plotted using the R package ‘semPlot’ [39].

## 3. Results

### 3.1. Descriptive Data

Table 2 provides the mean scores and standard deviations for the main outcome variables of valence, arousal, characters/narrative, music/audio visual and immersion.

### 3.2. Direct and Indirect Effects on Valence

Partial mediation model revealed that characters/narrative had a significant, yet rather small, positive indirect effect mediated by immersion upon affective valence (β_std_ = 0.122 [95%CI 0.013 to 0.231]; *p* = 0.012), but there was not a significant direct effect (β_std_ = −0.057 [95%CI −0.257 to 0.143]; *p* = 0.524) nor was the total effect significant (β_std_ = 0.066 [95%CI −0.109 to 0.241]; *p* = 0.401). Music/Audio-visual elements however had a significant positive direct effect possibly ranging from small to moderate (β_std_ = 0.449 [95%CI 0.264 to 0.634]; *p* < 0.001), and a significant though small positive indirect effect mediated by immersion (β_std_ = 0.122 [95%CI 0.014 to 0.229]; *p* = 0.011), resulting in a possible small to large total effect upon affective valence (β_std_ = 0.570 [95%CI 0.426 to 0.714]; *p* < 0.001). Figure 2 shows the path diagram for this partial mediation model, which can also be seen in Table 3.

### 3.3. Direct and Indirect Effects on Arousal

Partial mediation model did not reveal any significant effects for characters/narrative upon arousal. Direct (β_std_ = 0.041 [95%CI −0.203 to 0.285]; *p* = 0.376), indirect (β_std_ = 0.034 [95%CI −0.093 to 0.161]; *p* = 0.601), and total effects (β_std_ = 0.075 [95%CI −0.133 to 0.283]; *p* = 0.807) were all non-significant. Music/audio-visual elements however had a significant positive direct effect possibly ranging from small to almost moderate (β_std_ = 0.244 [95%CI 0.006 to 0.482]; *p* = 0.021) and a possible small to almost moderate total effect upon arousal (β_std_ = 0.278 [95%CI 0.078 to 0.477]; *p* = 0.002). There was however no significant indirect effect mediated by immersion (β_std_ = 0.034 [95%CI −0.092 to 0.160]; *p* = 0.548). Figure 3 shows the path diagram for this partial mediation model, which can also be seen in Table 3.

## 4. Discussion

This study aimed to investigate the influence of immersive elements on children’s post activity affective response to the Move Like the Avengers physical activity video experience. Overall, the post activity affective responses were positive, children reported feeling “good” and “awake” for valence (mean score: 3.6 ± 2.2) and arousal (mean score: 5.1 ± 1.0), respectively. Most interestingly the study also explored how the variables of characters/narrative and music/audio visual influenced the affective response through the mediating role of the sense of immersion. Results revealed for valence there was a significant, yet small, positive indirect effect mediated by immersion from characters/narrative. Additionally, for valence a significant positive direct effect, and a significant though small positive indirect effect mediated by immersion of the music/audio visual elements, resulting in a possible small to large total effect. This suggests that both the use of characters/narrative, and the music/audio visual variables helped to create a sense of immersion during the activities which resulted in more favourable valence scores. Direct, but not indirect (mediated by immersion) effects were also found for the arousal scores, specifically the music/audio visual elements which had a significant positive direct effect and a possible small to almost moderate total effect.

When narratives are used in games and activities this can help to increase engagement and one way this is achieved is through interactions with characters [15]. Often characters actions are purposely aligned to physical activity skills allowing for a synchronisation between the character and the one engaging in the activity [40]. This can be seen in the Move Like the Avengers Video where a number of Avenger characters are included and at different points certain actions are aligned to those of each character. For example, Figure 4a shows how Thor’s flying is likened to the instructors balancing, and in Figure 4b Black Widow’s fighting is likened to a martial art stance. This alignment of the characters moves to those of the child engaging in the video links back the research on wishful identification. This occurs when the viewer wants to be like the character, and they actively emulate the character who they look up to [17]. In this video the Avengers are representing the characters that children look up to and want to emulate and so, as the Avengers in the video are completing physical activity moves, children will want to engage in the physical activity too. One item included in the survey asked children to answer how much they agreed with the statement “I’d like to do the kinds of things the Avengers do”; in total 75% of the children agreed to some extent with this statement (32% agreed “lots”, 43% agreeing “a little bit”). This suggests there were levels of wishful identification present between the children included in this sample and the Avengers characters present in the video.

This wishful identification with characters has also been reported elsewhere in the literature around engaging children in physical activity. Reporting on the Change4Life 10-min Disney Shake ups, Public Health England [41] reported that 64% of children surveyed would be more physically active if they saw their favourite Disney characters being active. This was further supported in a recent study into these 10-min shake up activities, where the post activity affective response was reported, as well as qualitative comments from parents [28]. Qualitative data (e.g., “the use of characters was useful, they enjoyed having specific roles to play”) suggested that not only are characters effective at facilitating specific physical activity skills, but they can also lead to a more enjoyable experience. Characters may help to create a more enjoyable experience for children by providing someone to identify with, identification with a character allows the child watching to forget about themselves and assume the identity of the character [42]. This is often understood as a friendship and, as an emotional process where the child takes on the role of the character in the narrative, here happiness can result from events where the character meets their goals [42]. In the Move Like the Avengers video children could identify with the characters included and become an Avenger, completing their training to be like Earth’s mightiest heroes. This could lead to a more positive emotional response, especially if this also ties into the wishful identification processes. As they will want to be like the characters, acting like them will achieve this. This is supported by both the current study into the use of Marvel Avenger characters, and the previous study into multiple Disney characters, as both reported similar positive affective responses post activity (Avengers mean valence: 3.6, Avengers mean arousal: 5.1, Change4Life mean valence: 2.7 Change4life mean arousal: 4.6) This provides some justification for the incorporation of characters and narratives into a child’s physical activity experience. Indeed, this meets the recent calls in the literature for physical activity interventions to focus more on ensuring an enjoyable experience [4].

The use of characters in a narrative was found to influence the valence response of the children through mediation with a sense of immersion. As noted, children seem to enjoy activities where they are “caught up in the moment” and “only concerned with the activity and nothing else” [12]. Characters and narratives are often described as transportation tools used to engage the viewer affectively while simultaneously reducing their motivation for counterarguing; as they are transported into the narrative world emotional reactions are aroused [43]. The transportation is based on a metaphor as the one engaged undertakes a mental journey into the narrative where they become immersed in the story, and the characters are central components to the transportation and immersion [44,45]. Another way to immerse those engaged is through the music, which was also included in the Move like the Avengers video and investigated in this study. When used in a game or activity music can provide the background sounds, it can be used to direct attention to specific events, it can be a warning sign for danger, be used to indicate emotions and success and/or failure [18]. Music can also have an influence over the affective response to the session and has multiple benefits when used in a physical activity setting [19,46]. In this study the music used in the Move like the Avengers video was found to have an effect on the valence response of those children engaging which was partially mediated by a sense of immersion in the video, and also a direct effect on the arousal.

Music is often incorporated into exergames, a method of gameplay which involves body movements to move the game forward, and in addition to entertainment the game has a focus on enhancing physical activity [47]. Exergames have a particular focus on intrinsic motivation and have fun enhancing aspects [40]. The blending of music with the interactive nature of gameplay can provide an experience that draws players into the world of the game [48]. In a recent study Kegan [48] investigated the game “Crypt of the Necrodancer” where music is used as a controlling element that reinforces the sense of transportation and immersion in the game. In the game players have to move in time to the music to get to the next level. It was reported that the integrated nature of its immersive connections helps to foster a feeling of investment and flow. Further they suggest investigating the links between game play and musical interactivity to influence immersion might further understanding of how we engage with media at a deeper level.

What was novel however about this current study was the combined used of music and characters within a narrative to help increase the sense of immersion in the activity. As stated previously children seem to enjoy activities where they feel immersed, and it was reported in the current study that the elements of characters/narrative and music/audio visual mediated the affective valence scores through a sense of immersion. Research into immersive physical activities for children is limited, however there has been some relatable research conducted in game-based research. For example, a game experience is often described as a flow experience, where engagement is facilitated through the features of the game, this results in a playful experience [18]. Plass [18] provided a model of game-based learning where these game features are included, there are five in total: incentives, game mechanics, aesthetic design, narrative design and music score. The last two link back to the current study where characters in a narrative and music/audio-visual elements were included. It is unclear whether the influence on immersion is greater when more immersive elements are included, or whether there might be diminishing returns for each additional element. Further there may potentially be other additional elements that could be incorporated to help increase the immersive experience of the activity. This is something that future research should explore, as this study has shown initial benefits to the inclusions of immersive elements which echo’s previous descriptive research suggesting that children find these types of activities enjoyable [12].

### Limitations and Future Research

One limitation of the current study was the survey used. This survey was based on previous existing measures, some items of which had not been validated for use with children and further were adapted in the present application. We also did not conduct any psychometric evaluation of the items used instead relying on their face validity and thus an Intuitive Test Theory [37]. As this study was exploratory, the survey was used considering a balance between feasibility and information on the constructs being investigated and we felt facilitated some insight into immersive activities for children. However, future research should look to develop a more robust tool to evaluate these constructs in these populations, particularly now that they have been identified as a promising area of research.

This study was also limited to the online environment, so using digital versions of tools for the constructs being investigated as well as having no control over the exact engagement with the activity experience. While this perhaps added to the ecological validity of the intervention as it was performed in the environments it would typically be engaged with, it would be interesting to investigate this with participants in person to perhaps gather information on the actual intensities of effort during the activities, perhaps using heart rate or accelerometery measures. Further, whether a more “physical” version of similar activities might help create a more immersive atmosphere, or even a hybrid between the two where elements of the digital and physical environments are included, would be of interest to investigate. These limitations were however largely due to the safety measures currently in place due to the COVID-19 pandemic. For example, as the data had to be collected online the researchers could not be present to make sure that as soon as the video was completed the survey was answered, the participants were asked to complete the survey straight away after completing the video. However, without being physically present at data collection this could not be monitored. It is also worth noting that because of the timing of the data collection, being during a global pandemic, the emotions of the children surveyed may be affected by this. This is important to note as affective responses were collected and it is known that the pandemic has led to many mental health issues for children [49] so this should be considered when the results of this study are interpreted. However, the majority of children (~71%) reported that they maintained similar or increased levels of physical activity since before the pandemic, and also the majority (~91%) reported they were “Happy” or “Very Happy”, and “Not very worried” (~82%).

## 5. Conclusions

The aim of the research was to explore the influence of immersive elements on children’s post activity affective response to the Move like the Avengers physical activity video experience. The results suggest the use of characters within a narrative, and music/audio visual elements could help make physical activity enjoyable. Further, the study suggests these elements can help to create a sense of immersion during the activity, which can subsequently influence the affective response. As previous research has noted children find immersion in activities fun [12] and commented that future physical activity interventions in children should focus on promoting a more enjoyable experience [4] this study therefore represented an interesting addition to the literature. This exploratory study has provided an initial investigation exploring these elements in a physical activity setting with children, and future research should aim to investigate these further to gain a deeper and fuller understanding of their influence over engagement.

## Figures and Tables

**Figure 1 ijerph-18-07184-f001:**
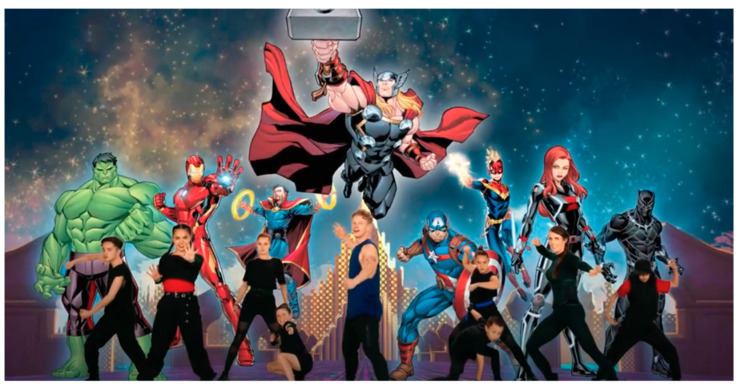
Move Like the Avengers Characters and Instructors.

**Figure 2 ijerph-18-07184-f002:**
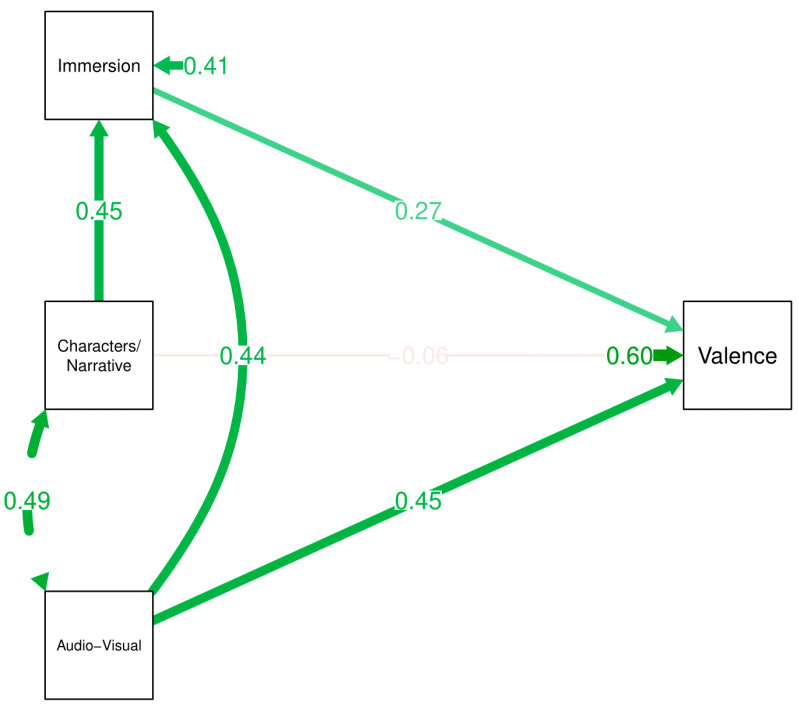
Path diagram for partial mediation model of affective valence. Note: aud = audio-visual; chr = characters and narrative; imm = immersion; vln = affective valence.

**Figure 3 ijerph-18-07184-f003:**
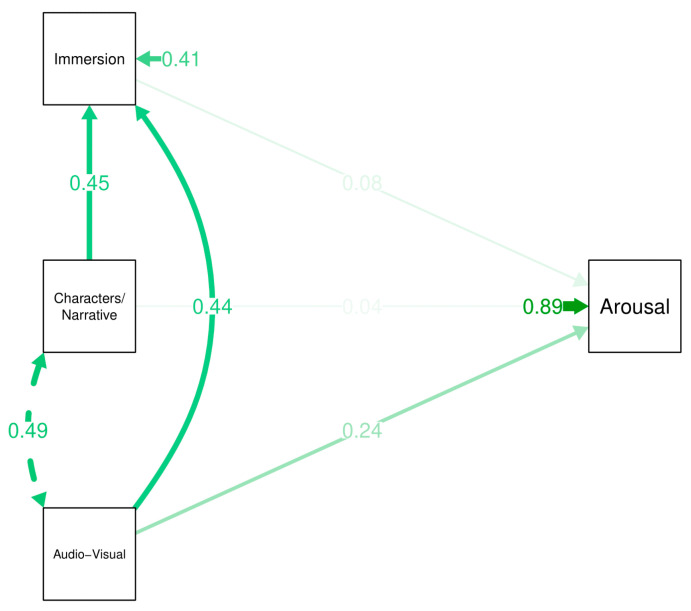
Path diagram for partial mediation model of arousal. Note: aud = audio-visual; chr = characters and narrative; imm = immersion; ars = arousal.

**Figure 4 ijerph-18-07184-f004:**
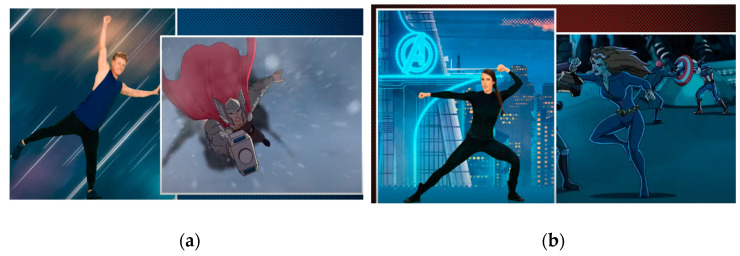
(**a**): Still from Video, Thor Flying, (**b**): Still from Video, Black Widow Fighting.

**Table 1 ijerph-18-07184-t001:** Participant Characteristics.

Characteristics	Sample
Gender	Boys (n = 65) Girls (n = 71)
Age	6 or younger (n = 18) Aged 7–11 (n = 116) 12 or older (n = 2)
Location	UK (n = 108) USA (n = 13) Oceania (n = 3) Europe (not UK) (n = 10) South America (n = 1) Asia (n = 1)
Location of video	Home (n = 70) School (n = 66)
Company	Alone (n = 42) With Friends/Family (n = 94)
Parent/guardian with membership	Yes (n = 30) No (n = 106)
Watched an Avengers Film	Yes (n = 79) No (n = 57)
Watched a film or read a comic with an Avengers character	Yes (n = 92) No (n = 44)
Completed the video previously	Yes (n = 42) No (n = 93)

**Table 2 ijerph-18-07184-t002:** Means and SD for Variables.

Variable	Mean ± SD	Scale Used
Valence	3.6 ± 2.2	11-point Likert scale: −5 (very bad) to +5 (very good)
Arousal	5.1 ± 1.0	6-point Likert scale: 1 (Very sleepy) to 6 (very wake)
Characters/Narrative	2.0 ± 0.7	3-point Likert scale (1–3)
Audio/Audio Visual	2.4 ± 0.6	3-point Likert scale (1–3)
Immersion	2.0 ± 0.4	3-point Likert scale (1–3)

**Table 3 ijerph-18-07184-t003:** Structural Equation Modelling Results.

	Valence			Arousal		
	β_std_	95% CI	*p*	β_std_	95% CI	*p*
Characters/Narrative						
Direct	−0.057	−0.257 to 0.143	0.524	0.041	−0.203 to 0.285	0.376
Indirect	0.122	0.013 to 0.231	0.012	0.034	0.093 to 0.161	0.601
Total	0.066	−0.109 to 0.241	0.401	0.075	0.133 to 0.283	0.807
Audio/Audio Visual						
Direct	0.449	0.264 to 0.634	<0.001	0.244	0.006 to 0.482	0.021
Indirect	0.122	0.014 to 0.229	0.011	0.034	0.092 to 0.160	0.548
Total	0.570	0.426 to 0.714	<0.001	0.278	0.078 to 0.477	0.002

## Data Availability

Pre-registration information available here: https://osf.io/4avtc (accessed on 30 June 2021), and Appendix A.

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
