# Peer review of "Can Earth’s Mightiest Heroes Help Children Be Physically Active? Exploring the Immersive Qualities of Les Mills’ and Marvel’s “Move Like the Avengers” Video"

_ijerph, 2021, doi:10.3390/ijerph18137184_

Round 1
Reviewer 1 Report
Introduction- well written. Potentially, it would be recommended to cut down.
Methods- 2.2 participants: most information should be under "Power calculation" section, not here. Line 138-144 would be appropriate in this section.
Statistical analysis: it was unclear to understand how the mediation analysis was conducted. Please include more information.
Results: Is it possible to include these data into a table? Also, a table for participants characteristics is missing.
Author Response
Thank you for your review and constructive comments on our manuscript. Please find attached the response to these.

Reviewer 2 Report
I thought this was an interesting study, and potentially very valuable in the design of appealing physical activity interventions aimed at children. It was well written, and the Discussion was very thorough. I only have two areas of concern (one may require extensive revisions, however):
Section 2.3 (Recruitment): How many schools were asked to participate in the study? Were the videos shown as part of a physical education class? If so, are you able to determine if the “group fitness” nature of the physical activity changed your results? Perhaps the children participating in the exercise as part of a group at school had a more positive affective response because they were exercising with friends and classmates, and the “group” nature of the physical activity was the largest influence on valence and arousal. This is my biggest concern. You essentially have two groups- those who did this activity at home, and those who did the activity at school. Did you analyze the results for these groups separately? It would be interesting and worthwhile to see if valence, arousal, and the models differed between those two groups.
Section 2.4:
Line 162-164: How was guardian consent obtained for the children participating at school?
Author Response

(The authors gave the same response as above.)

Round 2
Reviewer 2 Report
The authors have addressed my comments satisfactorily.